# Effect of Depth on Ultrasound Point Shear Wave Elastography in an Elasticity Phantom

Fahad F. Almutairi [1,2,3,*], Rawan Abdeen [1], Jaber Alyami [1,2,3] and Salahaden R. Sultan [1]

1   Department of Radiologic Sciences, Faculty of Applied Medical Sciences, King Abdulaziz University,
    Jeddah 21589, Saudi Arabia; aeabdeen@kau.edu.sa (R.A.); jhalyami@kau.edu.sa (J.A.);
    srsultan@kau.edu.sa (S.R.S.)
2   Animal House Unit, King Fahd Medical Research Center, King Abdulaziz University,
    Jeddah 21589, Saudi Arabia
3   Smart Medical Imaging Research Group, King Abdulaziz University, Jeddah 22252, Saudi Arabia
*   Correspondence: ffalmutairi@kau.edu.sa

**Abstract: Background:** Phantom studies are widely used to assess variability in measurements. This study aimed to assess the reliability and accuracy of point Shear Wave elastography (pSWE) measurements of an elasticity phantom. **Methods:** Measurements were obtained by an experienced certified clinical sonographer at three different depth levels in kPa, using a curvilinear 5-1MHz transducer of the EPIQ7 ultrasound imaging system. **Results:** A total of 180 pSWE measurements were obtained at three different depth levels (three cm, five cm, and seven cm) of the phantom background. The mean CV of pSWE was low at all depths (3 cm: 8.8%; 5 cm: 7%; 7 cm: 7.2%). There was a significant difference between measurements at depths of 3 cm vs. 7 cm (MD: $-0.85$, 95% CI $-1.5$, $-0.11$, $p = 0.024$) and measurements at depths 5 cm vs. 7 cm (MD: $-1.1$, 95% CI $-1.7$, $-0.47$, $p = 0.001$). An overestimation of mean pSWE measurements at a depth of 7 cm was noted compared to the manufacturer's value (2.7%, $p = 0.006$). **Conclusions:** Superficial phantom SWE measurements in this study had low variability compared to deep measurement. pSWE measurements at deep levels can be overestimated.

**Keywords:** ultrasound; point shear-wave elastography; pSWE; elasticity phantom





## 1. Introduction

There has been increasing interest recently in evaluating the elasticity of different tissues using elastography techniques. The ability to quantify changes in tissue stiffness using elasticity imaging techniques improves the diagnosis of diseases at early stages [1]. Ultrasound elastography is relatively inexpensive, provides real time measurements of tissue stiffness [2], and is currently being used in clinical practice for assessing the liver, breasts, prostates, thyroids, and the musculoskeletal system [3].

It has been reported that ultrasound shear wave elastography (SWE) is a reproducible imaging method for assessing tissue stiffness [4]. There are three approaches to perform SWE, Transient elastography (TE), Two-dimensional SWE (2D SWE) and point SWE (pSWE) [5]. pSWE is an elasticity estimation technique that produces a shear wave by applying acoustic radiation force impulse (ARFI) in a small region of interest (about 1 cm$^3$). This technique stimulates the tissue to provide a quantitative stiffness metric that corresponds to tissue elasticity [6]. Several studies have recently been conducted to investigate the variability of pSWE measurements [7–12], in which a variation in measurements has been reported [7,10,13–17].

In vivo and in vitro studies have been undertaken to investigate the effect of measurement depth measured by pSWE on many applications [16,18–20]. Discrepancies in pSWE measurements have been reported due to differences in a body organ, used probes, or sample sizes that have been examined. However, these discrepancies might provide

inaccurate findings and thus affect the reliability of diagnosis. Therefore, phantom studies could be ideal for evaluating the performance of US elastography in a known depth.

Attenuation of the ARFI and tracking waves can be considered as a source of variability in pSWE measurements. In addition, the attenuation of tissue could dampen the signals of the ultrasound as increasing the depth limits the accurate measurement of deeper tissue [5].

To the authors' knowledge, only a few studies have assessed the effect of depth on SWE measurement using commercially available phantoms. As a result, we aimed to assess the reliability and accuracy of pSWE measurements at different depths in an elasticity phantom.

## 2. Material and Methods

### 2.1. Study Design

EPIQ7 Philips Health Care ultrasound imaging systems were used to assess the variability and accuracy of pSWE measurements at different depths in Young's modulus (kilopascal, kPa) on an elasticity quality assurance phantom using a curvilinear 5-1MHz transducer. An experienced certified clinical sonographer with efficient training on ultrasound SWE and blinded to the reference stiffness value was asked to take pSWE measurements every two centimeters at three different depths in a single visit.

### 2.2. Phantom

pSWE measurements were performed on an elasticity quality assurance phantom model 049 produced by Computerized Imaging Reference Systems (Figure 1) [21]. The phantom is made from Zerdine® with acoustic properties comparable to human tissue [22] and consists of spherical lesions. The phantom background and lesions have an elasticity value which is known in Young's modulus (background elasticity value is 25 kPa, see Table 1).

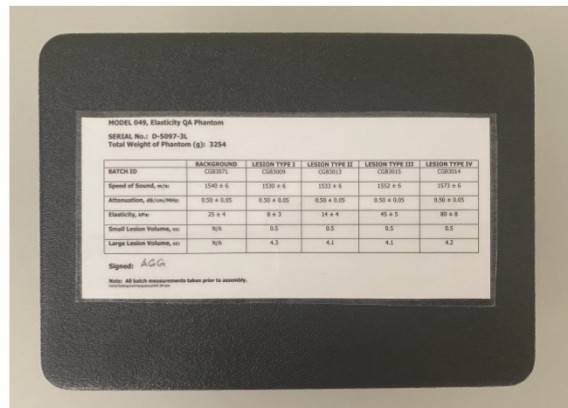 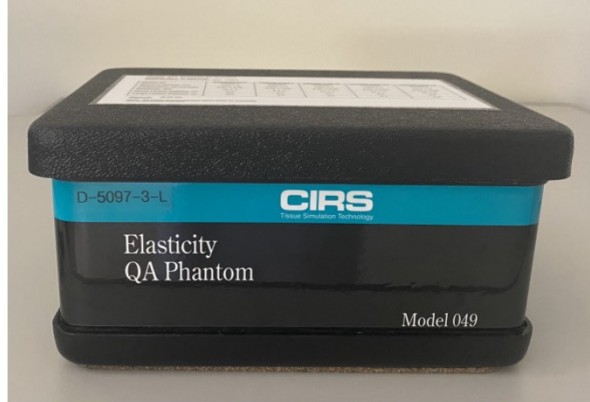

**Figure 1.** Computerized Imaging Reference Systems. Elasticity quality assurance phantom. Phantom Model-049. Norfolk, VA: Computerized Imaging Reference Systems, 2016. Available from: https://www.cirsinc.com/wp-content/uploads/2020/03/049-049-DS-030920.pdf (accessed on 10 May 2022) [21].

**Table 1.** Manufacture elasticity measurements of background and lesions within the phantom.

| Lesion Type | Young's Modulus (kPa, ±5% SD) |
|---|---|
| Background | 25 ± 4 |
| L I | 8 ± 3 |
| L II | 14 ± 4 |
| L III | 45 ± 5 |
| L VI | 80 ± 8 |

Abbreviation: kPa: kilopascal; SD: standard deviation.

### 2.3. Data Acquisition

The sonographer was asked to take 60 pSWE measurements at three different depth levels in kPa using a curvilinear 5-1MHz transducer of the EPIQ7 ultrasound imaging system. Due to the fact that the curvilinear 5-1MHz transducer provides low image quality at a superficial area, a level of three centimeters and deeper within the phantom was assessed in this study. The data were acquired from the phantom background at depths of three, five, and seven centimeters (i.e., the maximum depth in which pSWE measurements can be obtained with the use of a EPIQ7 ultrasound system, Figure 2) in longitudinal scanning with the transducer lifted and repositioned between acquisitions by applying minimal pressure on the phantom surface [23]. Ultrasound system controls were optimized pre-measurements. pSWE measurements were obtained using a medium penetration mode with a gain value of 71% in EPIQ7 and the image depth was set at 10 cm. The pSWE region of interest (ROI) was fixed and un-adjustable with an approximate diameter of one cm (Figure 2).

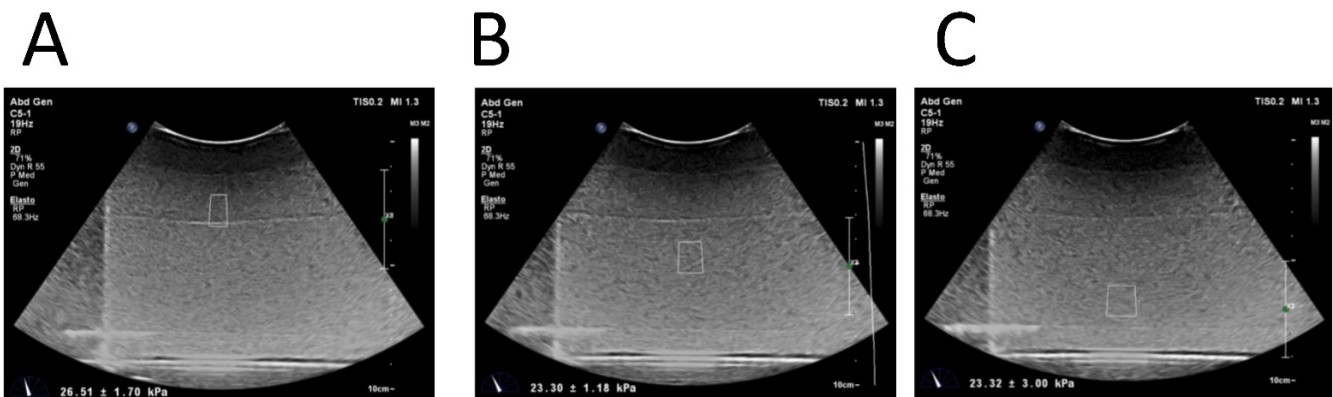

**Figure 2.** pSWE measurements of background within an elasticity quality assurance phantom (Model 049, CIRS). pSWE were obtained in kPa using EPIQ7 ultrasound imaging systems at different depths (i.e., three cm (**A**), five cm (**B**), and seven cm (**C**)) with the use of medium penetration, gain of 71% and image depth was set at 10 cm. The pSWE region of interest (ROI) was fixed and un-adjustable having an approximate diameter of one cm.

### 2.4. Statistical Analysis

The mean of coefficient of variation (CV) of pSWE from the phantom background was used to determine measurement variability at each depth; <15% low variability; 16–25% moderate variability; >25% high variability [24]. An independent sample *t*-test was used to compare the means of pSWE measurements at different depths. The accuracy of elasticity measurements from manufacturer values was determined using a one-sample *t*-test. The level of significance was set at <0.05. Additionally, a statistical analysis was performed using IBM SPSS Statistics and GraphPad PRISM 7.

## 3. Results

A total of 180 pSWE images with elasticity measurements in kPa were obtained. The sonographer also acquired 60 pSWE measurements from each depth level (i.e., three cm, five cm, and seven cm, Figure 2) of the phantom background.

### 3.1. Effect of Depth on pSWE Measurements

The mean of CV of pSWE was low at all depths (3 cm: 8.8%; 5 cm: 7%; 7 cm: 7.2%). Likewise, there was no significant difference between pSWE measurements at depths of 3 cm vs. 5 cm (mean difference (MD): 0.27, 95% confidant interval (CI) −0.43, 0.99, *p* = 0.44, Figure 3A). However, there was a significant difference between pSWE measurements at depths of 3 cm vs. 7 cm (MD: −0.85, 95% CI −1.5, −0.11, *p* = 0.024, Figure 3B) and measurements at depths 5 cm vs. 7 cm (MD: −1.1, 95% CI −1.7, −0.47, *p* = 0.001, Figure 3C).

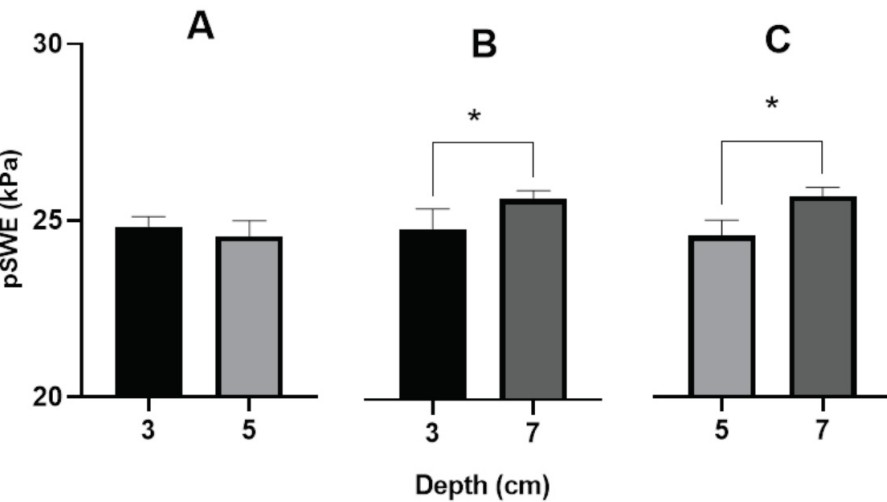

**Figure 3.** Comparison between pSWE measurements at different depths. Significant differences between pSWE measurements at depths of three cm vs. seven cm ($p = 0.024$, (**B**)) and between pSWE at depths five cm vs. seven cm ($p = 0.001$, (**C**)) were noted. No significant differences in pSWE at depths of three cm vs. five cm ($p = 0.44$, (**A**)). Mean, SEM (* $p < 0.05$ using Independent *t*-test between pSWE measurements at different depths, number of pSWE measurements per depth = 60).

*3.2. Accuracy of pSWE Measurements*

The pSWE mean at depths of 3 and 5 cm showed accurate measurements compared to the value reported by the manufactures (3 cm, $p = 0.44$; 5 cm, $p = 0.05$), but this is not the case at a depth of 7 cm, in which an overestimation of the pSWE mean compared to manufacture value was noted (2.7%, $p = 0.006$) (Figure 4).

**Accuracy of pSWE at different depths**

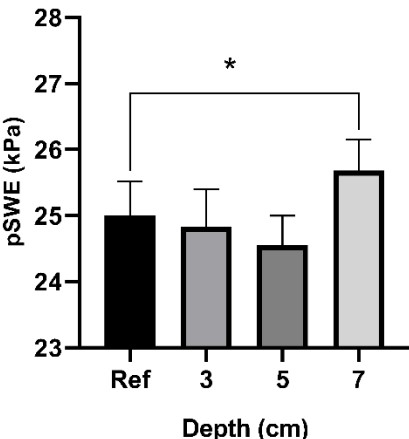

**Figure 4.** Accuracy of pSWE measurements at different depths compared to manufactrure reference value. There is a significant different between pSWE measurements at depths of seven cm compared to reference value ($p = 0.024$). No significant differences in pSWE at depths of three cm or five cm when compared to reference value. Mean, SEM (* $p < 0.05$ using one-sample *t*-test between pSWE measurements at different depths and reference value, number of pSWE measurements per depth = 60).

## 4. Discussion

The aim of this study was to assess the effects of depth on the variability and accuracy of pSWE measurements at different levels of depth in an elasticity phantom. The results showed that pSWE has low variability at all depths (i.e., three cm, five cm, and seven cm) with CV ranging from 7 to 8.8%. However, our results revealed significant differences in pSWE measurements at depths of seven centimeters when compared to those at depths of three and five centimeters. In addition, pSWE at depths of three and five cm showed more accurate measurements when compared to the value reported by the manufacturers, but the same is not true at a depth of seven cm. This suggests that pSWE at a deep level could significantly differ from pSWE measured at the superficial area and should be considered in future studies.

Similar findings to our study, which shows the variability in low pSWE measurements over different depth levels, have been previously reported [20], in which a low variability in measurement with CV of 6.8% was reported. However, in the aforementioned study, the depth level investigated was limited to a range between 1 and 4 cm. On the other hand, studies using more depth ranges of up to 6 cm have shown evidence for depth dependency with measurements [12,25,26]. In the previous studies, although the measurements were recorded in phantom settings, the shear wave technology used is different from the one in our study (2D-SWE vs. pSWE). This study also found no significant difference in pSWE at depths levels of three and five cm ($p = 0.44$). However, when depth levels increased, a significant difference was observed. pSWE measurements at depths of three cm and seven cm were significantly different ($p = 0.024$). Similarly, the variability significantly increased at depths of five cm and seven cm ($p < 0.001$). These findings concurred with what has been reported in other studies in terms of the remarkable effect of depth on elasticity measurements [12–15,18,26]. These results would therefore suggest that pSWE measurements should be taken at a depth of no more than 4 cm, as beyond this depth, the variability would significantly increase. This suggestion is also in line with the recent guidelines and recommendations on thyroid SWE, which emphasize that the depth of acquisition should be between 4–5 cm [27].

The accuracy of pSWE measurements over different depth levels was also investigated in this study. pSWE measurements were accurate for depths of three and five cm, in which no significant differences were noted when compared to the manufactures' value. However, the effect of depth on measurements becomes obvious as the depth increases to seven cm, where overestimated measurements were observed. A possible explanation for this discrepancy in measurement over depth is the technical limitation of the transducer in terms of ROIs. The scanner used in this study has a default ROIs shape and size which cannot be adjusted manually. Therefore, ROIs increase simultaneously with depth.

The current study has several limitations. pSWE measurements were obtained using a curvilinear transducer with a maximum depth of seven cm and, thus, deeper areas of more than seven cm were not assessed in this study. Likewise, pSWE measurements were taken from the background of phantom rather than spherical lesions; this was due to the poor image resolution with the use of a curvilinear transcoder at a depth of less than three cm in which the superficial lesions are placed (superficial lesions are at a depth of 1.5 cm). Furthermore, our ROIs were limited and constrained by the transducer's capability, which could have a negative effect on the depth measurements. Having another scanner with adjustable ROIs such as axipolere supersonic would overcome this limitation. Lastly, it is known that different factors have been reported to influence the measurements of pSWE. A recent paper published by our group investigated the accuracy and reproducibility of pSWE measured by different operators and ultrasound scanners [28] as factors that could affect pSWE, but not depth. For this reason, in this study, we focused on depth only as a source of variability in pSWE measurements.

## 5. Conclusions

This study showed that pSWE has low variability in measurements over depth ranges from three to seven centimeters in an elasticity phantom. However, pSWE measurements at a depth of seven cm and greater may differ significantly from measurements taken superficially (i.e., 5 cm or less).

## 6. Recommendation

Further studies assessing the effect of depth on different techniques of ultrasound shear-wave elastography using different transducers and scanners are required.

**Author Contributions:** F.F.A. and S.R.S. conceptualized and designed the study, and collected and organized data. R.A. and J.A. helped in organizing the collected data. S.R.S. analyzed and interpreted data. F.F.A., R.A., J.A. and S.R.S. wrote the initial and final draft of article as well as provided logistic support. All authors have read and agreed to the published version of the manuscript.

**Funding:** This research work was funded by the Institutional Fund Projects under grant no. (IFFPP-113-22). Therefore, the authors gratefully acknowledge technical and financial support from the Ministry of Education and the Deanship of Scientific Research (DSR), King Abdulaziz University, Jeddah, Saudi Arabia.

**Institutional Review Board Statement:** Not applicable.

**Informed Consent Statement:** Not applicable.

**Data Availability Statement:** The data that support the findings of this study are available from the corresponding author upon reasonable request.

**Conflicts of Interest:** The authors declare no conflict of interest.

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
