# Peer review of "Effect of Depth on Ultrasound Point Shear Wave Elastography in an Elasticity Phantom"

_applsci, doi:10.3390/app12136295_

Round 1

Reviewer 1 Report

Effect of Depth on Ultrasound Point Shear Wave 2 Elastography in an Elasticity Phantom

In the present study, the authors evaluated the reliability and precision of measurements of pSWE (elasticity estimation technique) at different depths (3.5 and 7 cm) in an elasticity phantom.

Overall, the work is interesting, but some drawbacks need to be improved before the work is published.

First, the authors could provide a succinct abstract for the manuscript.

However, even more, important and also highlighted by the authors themselves, "the present study has several limitations":

·         The pSWE measurements were obtained using a curvilinear transducer with a maximum depth of 7 cm. Therefore, a deeper area of more than 7 cm was not evaluated in this study.

·         pSWE measurements were taken from the background of phantom rather than spherical lesions.

·         Regions of interest (ROIs) were limited by the capacity of the transducer, which can harm depth measurements.

·         Although the authors focused their work on depth, other possible factors that could affect the measurements should have been evaluated.

Author Response

Manuscript ID: applsci-1756478

Title: Effect of Depth on Ultrasound Point Shear Wave Elastography in an Elasticity Phantom

Thank you for your time and expertise in reviewing the manuscript, which we have amended and hope you find much improved. Kindly find below responses to your comments. 

Reviewer Comments

Comment: Moderate English changes required 

Response:  The English language has been revised as suggested by a certified company with an editorial certificate.

Reviewer 1: In the present study, the authors evaluated the reliability and precision of measurements of pSWE (elasticity estimation technique) at different depths (3.5 and 7 cm) in an elasticity phantom. Overall, the work is interesting, but some drawbacks need to be improved before the work is published.

Comment: First, the authors could provide a succinct abstract for the manuscript.

Response: An abstract is provided to the manuscript

Comment: However, even more, important and also highlighted by the authors themselves, "the present study has several limitations":

  • The pSWE measurements were obtained using a curvilinear transducer with a maximum depth of 7 cm. Therefore, a deeper area of more than 7 cm was not evaluated in this study.

Response: Thank you for your comment. The maximum depth in which pSWE measurements can be obtained with the use of EPIQ7 ultrasound system is 7 cm. This considers as a technical limitation in EPIQ7 ultrasound system. This was added in lines 80-81. 

  • pSWE measurements were taken from the background of phantom rather than spherical lesions.

Response: The phantom contains spherical lesions at superficial depths (i.e.1.5 and 3 cm) with different elasticity values. Curvilinear transducer 5-1MHz provides poor image resolution at a superficial depth of 1.5 cm making it difficult to accurately measure pSWE at that level of depth. This was added in lines 76-78 and 171-175.

  • Regions of interest (ROIs) were limited by the capacity of the transducer, which can harm depth measurements.

Response: The scanner used in this study has a default ROIs shape and size which cannot be adjusted manually which also considered as a limitation of the ultrasound scanner used in this study, suggesting that using another scanner with adjustable ROIs would overcome this limitation. This was added in lines 178-179.

  • Although the authors focused their work on depth, other possible factors that could affect the measurements should have been evaluated.

Response: Thanks for your comment. Yes, we agree. “A recent paper published by our group has investigated the accuracy and reproducibility of pSWE measured by different operators and ultrasound scanners29 as factors that could affect pSWE, but not depth. For this reason, in this study we focused only on depth as a potential factor that may affect pSWE measurements”.  This was added in lines 180-182.

Reviewer 2 Report

Summary:

In this study, pSWE will be examined in more detail with regard to the depth of investigation. This will be done on a model.

General:

SWE is a new diagnostic method, which is also increasingly being investigated in neurology for muscle and nerve diseases. For the exact background of the application of this technology, the authors would like to get more information about possible influences during the examination by means of a model. However, some questions arise in the context of the application of a model. These are explained below.

- Introduction:

Unfortunate sentence: " however, the displacement of tissue itself is not measured" (line 29/30).

Please rephrase. In the end, it is about the interaction of waves with a tissue (in the context of which a "displacement" occurs) and what happens with it. The assumptions about this interaction should help to understand the stiffness. Therefore, please find a better description of the background.

- Methodology:

o The study was conducted with an "Experienced certified clinical sonographer". Who is this? Is this one of the co-authors? What is the exact job title? Is this a medical doctor? If so, what year of training/specialist/etc.?

o SWE is investigator-dependent. In the case of a theroetic question, which should fundamentally contribute to a better understanding, a comparison with an equally practised other songrapher, in the best case several education, could be useful. This would underline the observed effect and reduce the influence of factors such as more/less pressure when measuring.

o According to the authors, it seems to be known that the size of the ROI depends on the measurement depth. Here, it is recommended to use a device with a constant measured value. An influence on the values is conceivable.

- The criticism regarding the results and methods is mainly due to the methodology. This should be taken into account accordingly in both parts.

Author Response

Manuscript ID: applsci-1756478

Title: Effect of Depth on Ultrasound Point Shear Wave Elastography in an Elasticity Phantom

Thank you for your time and expertise in reviewing the manuscript, which we have amended and hope you find much improved. Kindly find below responses to your comments. 

Reviewer Comments

Reviewer 2: In this study, pSWE will be examined in more detail with regard to the depth of investigation. This will be done on a model. SWE is a new diagnostic method, which is also increasingly being investigated in neurology for muscle and nerve diseases. For the exact background of the application of this technology, the authors would like to get more information about possible influences during the examination by means of a model. However, some questions arise in the context of the application of a model. These are explained below.

 Comment: Introduction

Unfortunate sentence: " however, the displacement of tissue itself is not measured" (line 29/30). Please rephrase. In the end, it is about the interaction of waves with a tissue (in the context of which a "displacement" occurs) and what happens with it. The assumptions about this interaction should help to understand the stiffness. Therefore, please find a better description of the background.

Response:  We agreed on this comment and preferred to remove it from the text.

Comment: Methodology

  • The study was conducted with an "Experienced certified clinical sonographer". Who is this? Is this one of the co-authors? What is the exact job title? Is this a medical doctor? If so, what year of training/specialist/etc.?

Response: Yes, one of the co-authors (F.F.A.) his job title is an assistant professor of medical ultrasound with a PhD in ultrasound shear wave elastography, and certified by the Saudi Commission for Health Specialists as a senior specialist of medical ultrasound with a clinical experience of more than 8 years. 

  • SWE is investigator-dependent. In the case of a theoretic question, which should fundamentally contribute to a better understanding, a comparison with an equally practised other sonographer, in the best case several education, could be useful. This would underline the observed effect and reduce the influence of factors such as more/less pressure when measuring.

Response: Thank you for this comment. This is an interesting question to answer in future work. From our point of view, proper education (not necessary an academic degree; could be an intensive short course on the physics principle of elastography) and efficient training in using ultrasound elastography for the assessment of tissue elasticity is crucial to determine accurate measurements and would provide the operator with confident in using elastography as an assessment tool.  

  • According to the authors, it seems to be known that the size of the ROI depends on the measurement depth. Here, it is recommended to use a device with a constant measured value. An influence on the values is conceivable.

Response: We noticed that there is an increase in ROI at a deep level when compared to the superficial area size. The scanner used in this study has a default ROI shape and size which cannot be adjusted manually which consider a limitation of the ultrasound scanner used in this study. This was added in lines 178-179.

Comment:  The criticism regarding the results and methods is mainly due to the methodology. This should be taken into account accordingly in both parts.

Response: Thank you for your comments.  

Round 2

Reviewer 1 Report

Overall, the quality of the manuscript improved significantly after the revisions.